

# Variation in purple sea urchin (*Strongylocentrotus purpuratus*) morphological traits in relation to resource availability

Joshua G. Smith and Sabrina C. Garcia

Department of Ecology and Evolutionary Biology, University of California, Santa Cruz, Santa Cruz, California, United States

## ABSTRACT

Flexible resource investment is a risk sensitive reproductive strategy where individuals trade resources spent on reproduction for basic metabolic maintenance and survival. This study examined morphological variation in herbivorous sea urchin grazers across a mosaic landscape of macroalgae dominated habitats interspersed with patches of sea urchin barrens to determine whether sea urchins shift energy allocation in response to food limitation. Extensive underwater surveys of habitat attributes (e.g., sea urchin density, algae cover) were paired with detailed laboratory assays (e.g., sea urchin dissections) to determine how resource abundance affects energy allocation between reproductive capacity and body structure in the purple sea urchin, *Strongylocentrotus purpuratus*. We found that: (1) sea urchins had a more elongate jaw structure relative to body size in habitats void of macroalgae (i.e., barrens), (2) sea urchin reproductive capacity (i.e., gonad index) was lower in barrens and the barrens habitat was primarily comprised of encrusting algae, and (3) sea urchin jaw morphology (i.e., lantern index) and reproductive capacity (i.e., gonad index) were inversely related. These results suggest that sea urchins respond to macroalgae limited environments by shifting energy allocation between reproductive capacity and modifications of the foraging apparatus, which may explain the ability of sea urchins to acquire food in resource-limited environments.

# INTRODUCTION

A central issue in life history theory is whether (and how) organisms can be flexible in how they invest resources (*Bradshaw, 1965*; *Bårdsen et al., 2011*). When faced with food limitations, many organisms will reallocate internal resources, shifting from reproduction and energy storage to basic metabolic maintenance (*Braby & Jones, 1995*; *Fiore & Rodman, 2001*; *Bauerfeind & Fischer, 2005*). This is especially true in dynamic environments where organisms are susceptible to rapid (or even seasonal) changes in resources, or experience periods of extreme climatic events (*Boggs & Ross, 1993*; *Lau et al., 2009*). Therefore, understanding the spatial and temporal scales over which shifting responses occur can provide insight into when, where, and under what conditions organisms exhibit

Corresponding author
Joshua G. Smith, JogSmith@ucsc.edu

the plasticity to allocate resources to favor survival, or maintain their reproductive capacity.

Sea urchins (Echinoidea) are important benthic herbivores in many marine systems because of their ability to fundamentally transform macroalgal-dominated communities to 'barrens' void of macroalgae (*Filbee-Dexter & Scheibling, 2014*). The ability for sea urchins to reallocate resources may be one explanation for the persistence of dense aggregations of sea urchins in barren habitats with little food availability (*Simenstad, Estes & Kenyon, 1978*; *McShane & Anderson, 1997*). Several studies have documented variation in sea urchin jaw structure and body shape as a result of changes in food availability (*Ebert, 1980*; *Black, Johnson & Trendall, 1982*; *Black et al., 1984*; *Levitan, 1989*, *1991*). Food stress can also negatively impact reproductive success. Sea urchin gonads are both reproductive organs and important energy reserves that serve as a proxy for measuring overall health (*Walker, 1981*; *Lawrence & Lane, 1982*; *Rogers-Bennett et al., 1995*). Gonadal condition has been found to decrease over extended periods of food limitation as sea urchins are forced to expend energy reserves as food supplements (*Vadas, 1977*; *Andrew, 1989*; *James, 2007*; *Schroeter et al., 2009*). While these studies suggest that sea urchins possess a high degree of phenotypic plasticity (i.e., overall test size and jaw structure), the spatio-temporal scale over which population-level morphological responses occur is less understood and may only be revealed by tracking population responses through significant environmental changes, such as modifications in the abundance of macroalgae.

Along the central coast of California, active grazing by purple sea urchins (*Strongylocentrotus purpuratus*) shifted expansive kelp forests into a patchy mosaic of remnant kelp forests interspersed with sea urchin barrens (*Smith et al., 2021*). Beginning in the year 2014, an outbreak of purple sea urchins initiated kelp deforestation, which provided a unique and timely opportunity to explore sea urchin phenotypic plasticity in a recently transitioned system. Kelp forests and sea urchin barrens represent two important grazing environments. In patches of kelp forests, sea urchins are often at low densities and primarily employ a risk-averse passive-grazing strategy on detrital (hereafter, 'drift') kelp (*Filbee-Dexter & Scheibling, 2014*). However, in sea urchin barrens where drift is limited, sea urchins alter their behavior, emerge from the refuge space of crevices, and actively graze at high densities on nutrient-poor encrusting algae species (*Harrold & Pearse, 1987*; *Steneck et al., 2002*; *DeVries, Webb & Taylor, 2019*). Therefore, the success and survival of sea urchins in these two distinct habitats may be strongly dependent on the ability of sea urchins to modify resource allocation from reproduction (i.e., gonad development) to survival (i.e., metabolic maintenance).

Using the mosaic landscape of kelp forests and sea urchin barrens that initiated just 3 years prior to this study, we examined morphological variation in three fundamental sea urchin traits: jaw shape, body shape, and gonad condition. Specifically, we tested the following hypotheses: (1) sea urchin jaw (hereafter, Aristotle's 'lantern') morphology varies as a function of habitat type (i.e., kelp forest, sea urchin barren), (2) differences in observed lantern morphology between habitat types is a function of the algal assemblage in each of those habitats, and (3) sea urchin reproductive capacity (i.e., gonadal condition) is

inversely related to lantern morphology, indicating resource trade-offs between reproduction and metabolic maintenance.

## METHODS

We combine spatially explicit observations of habitat attributes (algae cover, sea urchin density) between kelp forests and sea urchin barrens with detailed measurements of morphological traits (jaw shape, body shape, gonad condition) of sea urchins collected from the field to determine the consequences of resource variation on sea urchin morphology.

### Survey design

This study was conducted along the southern end of Monterey Bay, California, USA. In 2014, overgrazing by purple sea urchins shifted a once expansive kelp forest landscape to a patchy mosaic of remnant forests interspersed with sea urchin barrens (*Smith et al., 2021*). While this study was conducted just 3 years after the initiation of the urchin outbreak, the spatial extent of barren patches continued to expand both during and for several years following this study.

Underwater surveys were conducted from June to September in 2017. A total of 83 randomly assigned subtidal survey sites within the study region were sampled to evaluate sea urchin size structure and density, habitat type (i.e., barrens, kelp forests), and algae cover (Fig. 1). Survey sites were located between the Breakwater (36.61237578 N, 121.8947703 W) and Point Pinos (36.63878914 N, 121.9246315 W) in Monterey, CA, in 5 to 20 m of water on rocky reef substratum. Each site was categorically assigned in situ as one of two habitat types: kelp forest (characterized by high macroalgae presence and low sea urchin density), or sea urchin barren (characterized by low macroalgae presence, dominated by coralline algae, and with high sea urchin density). These two categories formed the basis for all categorical comparisons and analyses.

Each site was surveyed using a radial sampling design following the methods described in *Smith et al. (2021)*. Briefly, a total of eight 5-m long transects were surveyed at each site, radiating from a fixed central position. A single transect was assigned to each cardinal (N, S, E, W) and inter-cardinal (NE, NW, SE, SW) direction around the survey site. Two randomly placed 1m x 1m photographic quadrats were sampled per transect (16 total quadrats per site). The location of the quadrats along the transects were assigned using a randomly stratified design so that the quadrats were not intentionally biased towards either the center or outer edge of the site. Within each quadrat, the total number of sea urchins were quantified by searching in crevices and carefully examining dense algae. By utilizing the same sampling design at each site, each of our 83 survey sites represent a replicate sample.

### Variation in sea urchin morphology between habitats

To test the hypothesis that lantern morphology and body shape varies as a function of habitat type, we conducted a series of laboratory dissections with sea urchins collected from spatially referenced field survey sites (study animal collection approved by the

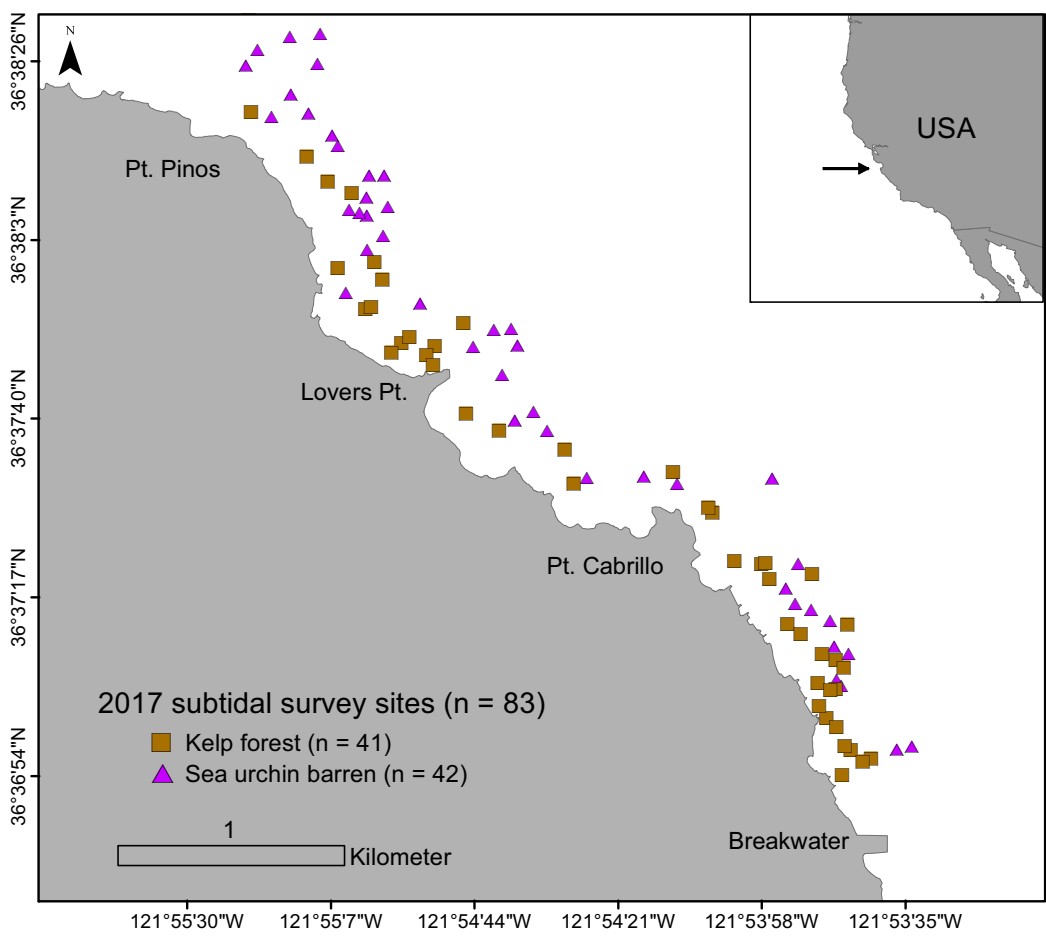

**Figure 1 Subtidal survey sites along the Monterey Peninsula, California, USA.** Brown squares indicate kelp forests and purple triangles indicate sea urchin barrens.

California Department of Fish and Wildlife, permit no. SC-389). Two sea urchins were randomly collected from fixed opposite corners of each quadrat for a maximum of 32 sea urchins per site. Only urchins greater than 3 cm were collected for dissections. These size classes of urchins are too large to have settled from the plankton later than 2014. Therefore, most of the dissected urchins were likely present during the 2014 habitat transformation event.

A total of 1,058 purple sea urchins were collected for this study. Upon collection, sea urchins were immediately 'fixed' following the methods of *Harrold & Reed (1985)* to preserve gonads for dissection. Sea urchins were injected with 3–15 mL of 10% neutral-buffered formalin (depending on animal size) through the peristomal membrane. After a 24-h fixation period, sea urchin tests were measured to the nearest 0.5 mm using Vernier calipers. Test diameter was recorded as the maximum distance across the test, excluding spines, and test height was measured as the length between the oral and aboral surfaces (to the nearest 0.05 mm). After these measurements were obtained, sea urchins were dissected for gonads and to measure lantern morphology. Our measure of lantern

length was recorded as the distance from the oral tip to the aboral surface of the epiphysis. Lantern width was measured as the maximum diameter of the aboral epiphysis surface.

We evaluated both sea urchin lantern length and gonad weight as independent functions of test diameter between habitat types (barrens, forest) using separate analysis of covariance (ANCOVA) tests. In both models, we tested the homogeneity of slopes assumption by evaluating the effect of test diameter, habitat type (barren, forest), and their interaction. We found that the interaction was not significant, therefore we ran reduced two-term ANCOVAs with test diameter and habitat type as independent model terms.

Sea urchin gonad weight serves as a useful metric for assessing animal health and reproductive capacity (*Walker, 1981*; *Lawrence & Lane, 1982*). To account for the relationship between urchin size and gonad weight, we used a gonad index to comparatively evaluate sea urchin gonad condition and lantern morphology. Sea urchin gonads were removed from each individual, blotted dry, and wet-weighed to the nearest 0.001 g. Gonad index was calculated as the percent body mass comprised of gonads:

$$Gonad\ Index\ (GI) = \frac{Gonad\ Wet\ Weight\ (g)}{(Total\ Wet\ Weight\ (g))} \times 100 \tag{1}$$

Therefore, higher GI values indicate greater gonad mass relative to the animal's total body mass.

## Relationship between sea urchin morphology and algae cover

Photographic quadrats were used to estimate algae cover in order to test for a relationship between algae cover, sea urchin morphology, and condition. Within each quadrat, still imagery was collected using a downward facing GoPro Hero4 camera and two LED Sola video lights. Still photos of each quadrat were analyzed in ImageJ using 16 uniform-points per quadrat. Photos were analyzed in the lab where an observer recorded the organism below each point to the lowest taxonomic level possible. However, because many species of algae are difficult to taxonomically identify from photo imagery, algae were organized into the following morphological groups: brown algae (predominately *Macrocystis pyrifera*), foliose red algae (predominately *Chondracanthus* spp.), and encrusting algae (includes encrusting red and encrusting coralline algae). These morphological groups were selected based on their known status as indicators for kelp forests (brown algae, foliose red) or sea urchin barren (encrusting algae) habitats (*Filbee-Dexter & Scheibling, 2014*). First, we used a series of linear regressions to test for associations between the cover of algae and two metrics of sea urchin morphology (lantern index, gonad index). Because lantern length was directly proportional to test diameter in both habitats, we used a lantern index to account for length-size relationships (e.g., larger animals have larger lanterns), defined as:

$$Lantern\ Index\ (LI) = \frac{Lantern\ Length\ (mm)}{Test\ Diameter\ (mm)} \tag{2}$$

Therefore, higher LI values indicate a longer jaw length relative to body size. We then used a beta regression on the pooled quadrat data for each site to account for the
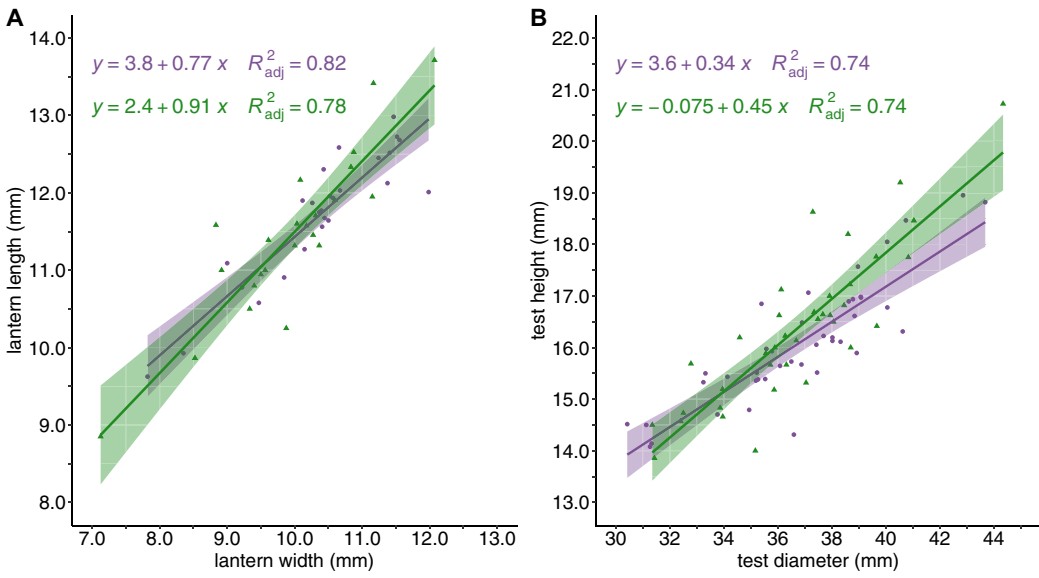

**Figure 2 Sea urchin shape and structure.** Sea urchin shape and body structure with (A) lantern length (mm) plotted against lantern width (mm), and (B) test height (mm) plotted against test diameter (mm). Linear regressions lines are shown for barrens (purple) and forests (green) and shaded with the 95% confidence intervals surrounding each mean.

proportional (0–1) response variable (*Ferrari & Cribari-Neto, 2004*), with mean sea urchin density as a continuous predictor. We examined sea urchin body size as the ratio of test diameter to test height between habitat types and found no significant difference. Additionally, lantern width was relatively proportional to body size.

## RESULTS

### Variation in sea urchin morphometrics as a function of habitat

Sea urchin density was greater in barrens ($M = 13.79 \pm 1.42$ S.E. individuals/m$^2$) than in the kelp forest habitat ($M = 5.33 \pm 0.69$ S.E. individuals/m$^2$, $P < 0.0001$). Although there were differences in sea urchin density between habitat types, the mean test diameter for sea urchins from barrens ($M = 36.58 \pm 0.51$ S.E. mm) and kelp forests ($M = 37.07 \pm 0.62$ S.E. mm) was not significantly different. Similarly, the relationships between lantern length and lantern width, and between test diameter and test height, did not vary with habitat type (Fig. 2).

An analysis of covariance (ANCOVA) revealed that relative lantern length was significantly greater in sea urchin barrens than in kelp forest habitats ($F = 72.63$, DF = 2 and 80, $P < 0.0001$). This difference in lantern length was most pronounced in individuals less than 49 mm (Fig. 3A). Although lantern length appeared to broadly converge between habitat types in individuals greater than 50 mm, the interaction between habitat type and test diameter was not significant. Conversely, gonad weight was significantly higher in the kelp forest habitat than in sea urchin barrens ($F = 76.54$, DF = 2 and 80, $P < 0.0001$; Fig. 3B).

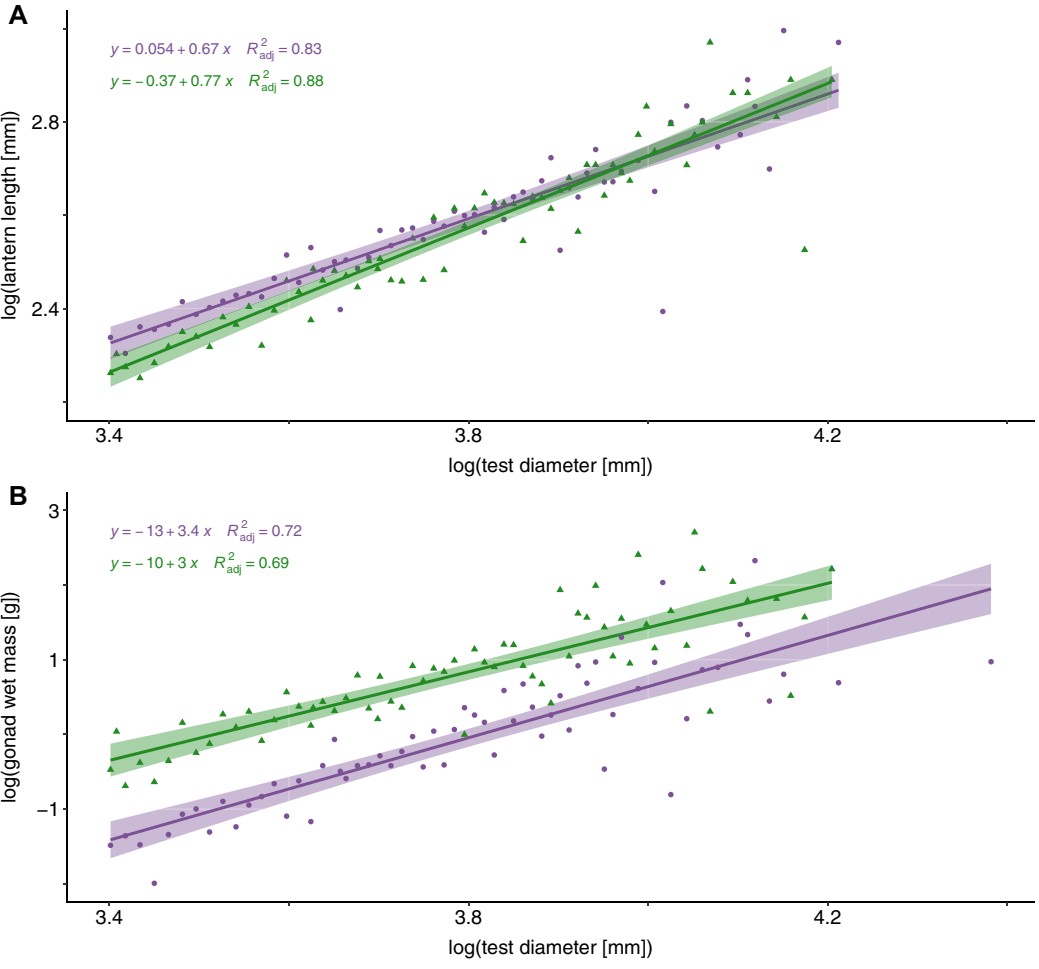

**Figure 3 Lantern length and gonad weight between habitat types.** Log-transformed lantern length (A) and gonad weight (B) between kelp forests (green) and sea urchin barrens (purple) with log-transformed test diameter as a covariate. Shaded regions depict the 95% confidence of fit surrounding each regression line.                                            

The percent cover of algae was correlated with both lantern index and gonad index (Fig. 4). There was a positive and linear relationship between encrusting algae and lantern index ($R^2 = 0.33$, $P < 0.001$; Fig. 4A), while brown ($R^2 = 0.14$, $P < 0.001$; Fig. 4C) and foliose red ($R^2 = 0.14$, $P < 0.001$; Fig. 4E) algae were negatively correlated with lantern index. In general, these lantern index relationships were inversely correlated with gonad index. Cover of encrusting algae was negatively correlated with gonad index ($R^2 = 0.4$, $P < 0.001$; Fig. 4B), while brown ($R^2 = 0.075$, $P < 0.02$; Fig. 4D) and foliose red ($R^2 = 0.35$, $P < 0.001$; Fig. 4F) algae were positively correlated.

## Habitat attributes of kelp forest and sea urchin barrens

We explored differences in the cover of key habitat-indicating algae groups (brown algae, foliose red, and encrusting algae) and sea urchin density between habitat types to determine factors that may influence variation in sea urchin morphological traits. A reduced two-term model with brown algae and red foliose algae explained the most

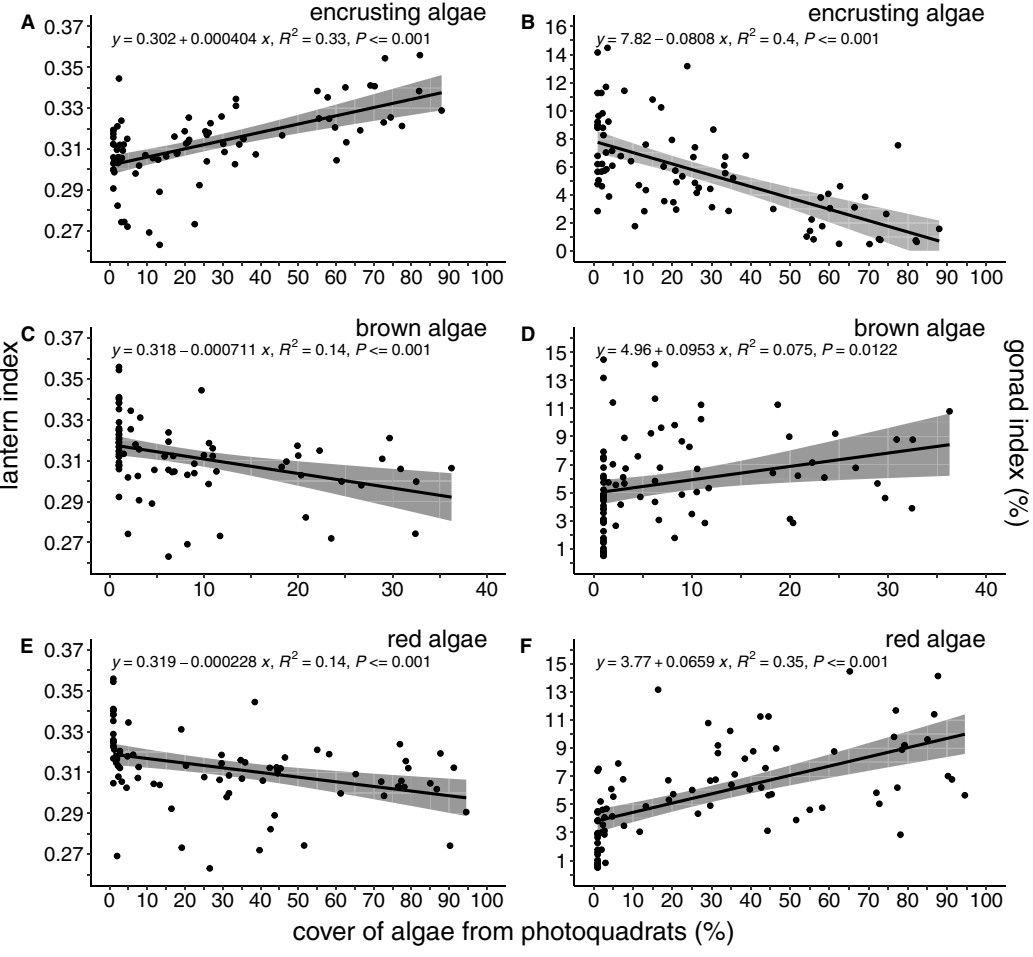

**Figure 4 Lantern index and gonad index as functions of the percent cover of algae.** Lantern index (A, C, E) and gonad index (B, D, F) as a function of the percent cover of encrusting (A, B), brown (C, D), and foliose red (E, F) algae. Each point depicts the mean cover of algae within a given site. Shaded regions indicate the 95% confidence of fit surrounding each linear regression line.

variation in habitat type (AIC = 61.48, DF = 3, P < 0.0001). Based on the model coefficients, high cover of brown and foliose algae explained a greater likelihood of a forest habitat, while the cover of encrusting algae was negatively correlated with the forest habitat. Because of the linear relationship between lantern index and sea urchin density, identical relationships were found between percent cover of algae and sea urchin density. Finally, there was a positive relationship between encrusting algae and sea urchin density (F = 36.54, P < 0.0001), while both the cover of brown (F = 12.47, P = 0.0007) and foliose red (F = 14.02, P < 0.0003) were negatively correlated with sea urchin density (Fig. 5).

# DISCUSSION

This study examined variation in sea urchin morphological traits (lantern shape, body shape, gonad condition) across a patchy mosaic landscape of kelp forests interspersed with

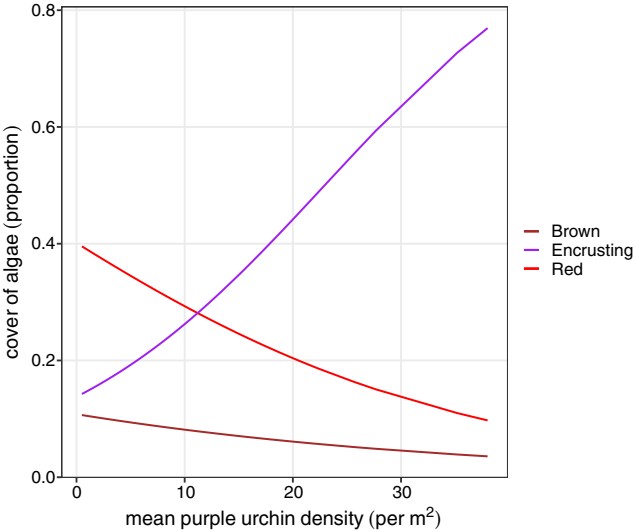

**Figure 5** Beta regression probability density functions for proportion of algae cover as a function of mean purple urchin density.

resource-limited sea urchin barrens. The inverse relationship between reproductive capacity (i.e., gonad index) and jaw morphology (i.e., lantern length) suggests that when food is scarce, lanterns are lengthened and gonad production declines. The cover of brown and foliose red algae was correlated with habitat type, which reinforces that key algal groups facilitate morphometric variation in sea urchins. Our findings reveal that sea urchins can alter the Aristotle's lantern to persist in dense aggregations and in low resource environments.

The sea urchin barrens that initiated 3-years prior to our study created a unique opportunity to explore variation in sea urchin morphology between barrens and macroalgae dominated habitats. This recently transitioned system comprised of a mosaic of kelp forests and sea urchin barrens helped to reveal that sea urchins may develop an enlarged feeding apparatus relative to body shape and size over a relatively short duration. This suggests that lantern morphology is likely influenced by the composition and abundance of macroalgae. Foliose red and brown algae are common within patches of kelp forest, while barrens are primarily dominated by encrusting coralline and encrusting red algae (*Simenstad, Estes & Kenyon, 1978*; *Filbee-Dexter & Scheibling, 2014*). One possible explanation for lengthened sea urchin lanterns in barrens is increased foraging ability, where a longer and more supported feeding apparatus enhances scraping efficiency for encrusting algae species that dominate sea urchin barrens. In our laboratory dissections, sea urchins with longer lanterns appeared to consume larger amounts of encrusting algae scraped from rocks than those with smaller lanterns. An increased ability to scrape food off rocks supports the positive relationship found in our study between encrusting algae and lantern index.

In addition to food limitation driving sea urchin lantern structure, sea urchins have also been shown to elicit similar morphological changes in response to differences in sea

urchin density (*Levitan, 1989*). Our study found a linear relationship between lantern index and sea urchin density, which is similar to that reported by *Black, Johnson & Trendall (1982)* in Rottnest Island, Western Australia with *Echinometra mathaei* having longer lanterns in areas of higher densities. Other field experiments have produced evidence for density dependent test growth and shrinkage (*Ebert, 1967*; *Pearse & Pearse, 1975*). Collectively, these studies highlight the dynamic nature of external factors that impact sea urchin morphology.

Gonad index is a metric often used to assess reproductive capacity, general health, and energy storage in sea urchins (*Walker, 1981*; *Lawrence & Lane, 1982*; *Ebert, Hernández & Russell, 2012*; *James & Siikavuopio, 2012*). In this study, gonad index was found to be greatest in forests and lowest in barrens, consistent with existing evidence that gonadal condition can be affected by the quantity and quality of available food (*Lau et al., 2009*). Sea urchins may also allocate energy based on their reproductive stage. For example, *Guillou, Lumingas & Michel (2000)* found that adult *Sphaerechinus granularis* fed in the pre-spawning stage only carried out gonadal growth and did not expend any resources to body growth, suggesting a seasonal energy trade off. This conversion has been found to happen within as short of a span as 3 weeks, suggesting that this rapid shift from gonadal development and nutrient storage to body maintenance is beneficial in a habitat rapidly transformed from food abundant to food limited (*Russell, 1998*).

This study explored variation in sea urchin morphometrics in response to the onset of widespread sea urchin barrens in Monterey Bay, California, USA. Our study reveals that sea urchins alter the structure of their foraging apparatus, Aristotle's lantern, and energy storage organs (gonads) without fundamental changes in body shape, and these changes may occur over a relatively short duration. This plastic morphological response to variation in food availability provides support that lengthened sea urchin lanterns may enhance grazing efficiency and success in food limited environments. Ultimately, these results highlight the ability for important kelp forest grazers such as sea urchins to persist in habitats that are void of macroalgae for extended periods of resource limitation.

## ACKNOWLEDGEMENTS

We deeply thank T. Gorra and dozens of volunteers who contributed to fieldwork and laboratory dissections. We also thank R. Mehta for initial feedback and the two reviewers whose comments substantially improved the manuscript.

### Funding

Funding for this project was provided by the National Science Foundation (NSF) Graduate Research Fellowship to Joshua G. Smith, the Future Leaders in Coastal Science Award to Joshua G. Smith and Sabrina C. Garcia, and by NSF Grant OCE-1538582. The funders had no role in study design, data collection and analysis, decision to publish, or preparation of the manuscript.

## Grant Disclosures

The following grant information was disclosed by the authors:
National Science Foundation (NSF).
Coastal Science Award.
NSF Grant: OCE-1538582.

## Competing Interests

The authors declare that they have no competing interests.

## Author Contributions

- Joshua G. Smith conceived and designed the experiments, performed the experiments, analyzed the data, prepared figures and/or tables, authored or reviewed drafts of the paper, and approved the final draft.
- Sabrina C. Garcia conceived and designed the experiments, performed the experiments, authored or reviewed drafts of the paper, and approved the final draft.

## Field Study Permissions

The following information was supplied relating to field study approvals (i.e., approving body and any reference numbers):

Study animal collection was approved by the California Department of Fish and Wildlife permit no. SC-389.

## Data Availability

The raw data and code is available in the Supplemental Files.

The data is also available at Dryad: Smith, Joshua; Garcia, Sabrina (2021), Variation in purple sea urchin (*Strongylocentrotus purpuratus*) morphological traits in relation to resource availability, Dryad, Dataset, DOI 10.7291/D11M38

## Supplemental Information

Supplemental information for this article can be found online at http://dx.doi.org/10.7717/peerj.11352#supplemental-information.

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
