# Peer review of "Variation in purple sea urchin (Strongylocentrotus purpuratus) morphological traits in relation to resource availability"

_PeerJ, doi:10.7717/peerj.11352_

## Round 0.1 · original submission · Major Revisions

Both reviewers made a number of thoughtful and constructive comments on your MS. Please address these, especially the suggestion that you graph the raw lantern and gonad data with body size as a covariate. This will work well for the categorical data (forest vs barrens), but I think you should also retain Fig 2 with the lantern index for the continuous data (% cover of algae) and present an analogous graph for the gonad index. It would also be useful to show the shape data (lantern length vs width, and body diameter vs height) in graph form, e.g., for the lantern data plot lantern length on the X, width on the Y, with points & regression lines for each of forest & barrens.

Please clarify your contribution of your study to the literature. In line 227 you state that "Our findings build on others (e.g., Ebert 1996 and 2014, Hughes 2012, McShane and Anderson 1997) by revealing that sea urchins may develop an enlarged feeding apparatus relative to body shape and size", but my understanding is that this is already well established. Is the strength of your study the large number of samples collected across a variable habitat, or the analysis of a recently changed habitat, or something else?

How old are your urchins likely to have been? I.e., for the ones in the barrens, what proportion of their lives was spent in this newly created habitat?

Line 180: what are the errors? SE?

Reviewer 1 ·

Basic reporting

The Literature Cited is poorly formatted and inconsistent. There also are errors in the Literature Cited. For example,
Station, B. M., Columbia, B. & Tg 1991. The title of the paper actually is for Levitan 1991. I checked the journal Ecology for 1991 with pages starting at 431 and there is no paper by Station et al.

Experimental design

Line 142. The lantern index. Why use an index? I looked at the data file and made a couple of graphs. It seems more appropriate to use an ANCOVA with test size as a covariate both for lanterns vs diameter and gonad vs weight. The limited range of measurements makes the results look linear and so analysis would be simple and, very important, graphs would be easier to visualize because actual data would be presented rather than index values.

Validity of the findings

no comment

Additional comments

Studies that are intensive or extensive and examined gonad yield need to be included. For example,
Schroeter, S. C., N. L. Gutiérrez, M. Robinson, R. Hilborn and Peter Halmay 2009. Moving from Data Poor to Data Rich: A Case Study of Community-Based Data Collection for the San Diego Red Sea Urchin Fishery, Marine and Coastal Fisheries: Dynamics, Management, and Ecosystem Science 1(1): 230-243.

Rogers-Bennett, L., W. A. Bennett, H. C. Fastenau and C. M. Dewees 1995. Spatial variation in red sea urchin reproduction and morphology: implications for harvest refugia. Ecol. Appl. 5:1171-1180.

Ebert, T. A., J. C. Hernández, and M. P. Russell 2012. Ocean conditions and bottom-up modifications of gonad development in the sea urchin Strongylocentrotus purpuratus over space and time. Marine Ecology Progress Series 467: 147–166.
Line 61-64. Change in jaw size associated with changes in food availability should include:
deVries, M. S., S. J. Webb, and J. R. A. Taylor 2019. Re‑examination of the effects of food abundance on jaw plasticity in purple sea urchins. Marine Biology 166:141 doi.org/10.1007/s00227-019-3586-1
deVries et al. argue that jaws are plastic but changes in relative jaw size are not adaptive.
Line 64. Check Dix (1972); he did not report that urchins could decrease in size. Also, decrease in body size has been reported as an artefact of measurement. See
Ebert, T. A. 2020. Growth and survival of postsettlement sea urchins. pp 95-145 In: J. M. Lawrence [ed.] Sea Urchins: Biology and Ecology 4rd edition (Developments in Aquaculture and Fisheries Science 38) Academic Press, San Diego, California.

Line 218. “reallocate” suggests that materials are taken from the gonads and used to build the lantern. A different view would be that when food is scarce gonads don’t grow and lanterns grow a bit better than the rest of the body.

Line 236. Urchins with larger lanterns scrape more algae off rocks is very interesting in light of the publication by DeVries et al. (2019) cited above. If there are quantitative data, and so more than just personal observation, this would make a very useful contribution because the only other works that actually measured scraping ability are those of Black et al. back in the ‘80s.

·

Basic reporting

Overall the manuscript is clearly written, however several consistency related issues need to be addressed:

- The use of P values for minor statistics is not consistent e.g. a P value is used when describing the results for sea urchin densities on line 157-158 but not for following results describing mean test diameter or body shape in lines 159-162. Similarly line 122 doesn’t not have an associated P value.

- The use of sea urchin and urchin is interchangeable throughout the manuscript. Suggest that sea urchin is used universally throughout.

- The use of barren, urchin barren and sea urchin barren is used throughout. Although all three are accepted terms it would be useful to decide upon one and use it consistently.

- Check that all species names are italisied in the Literature Cited section e.g. Line 283 and 293.

- Choose and or & between author names in text and in the Literature Cited section e.g. lines 324 and 334.

-Some journals titles are written in abbreviated form and some are not. Please be consistent throughout e.g. Lines 283 and 334.

- An M is used to denote mean values through the results section but is not used consistently e.g. it is used on Line 157 and 158 but not Line 159 and 160. Please also specify if the ± refers to standard error or standard deviation.

There are four in text references that do not appear in the Literature Cited section. Please either add to the Literature Cited section or remove if not relevant to the paper.
- Harrold and Pearse 1987 - Line 59.
- Ebert 1996 - Line 204.
- Leighton 1966 - Line 209.
- Lewis et al. 1990 - Line 222.

Line 29 - 31: This statement needs referencing.

Line 185: ...encrusting [algae] was negatively….

Line 202: You state that the barrens developed one year prior to the study. Should this not be three years? Lines 80 - 83 suggest that the barrens formed in 2014 and the experiments were conducted in 2017.

Figure 1: The right hand border for the inset box needs to be shifted slightly so it is obscured behind the main graph's border.

Experimental design

The research questions appear to fit within the primary research aims and scopes of the journal are well defined.

The most interesting aspect of this research to me is that it reports on a system that has so recently changed from kelp forest to urchin barren, therefore providing insight into what can be expected in terms of sea urchin phenotypic plasticity over the relatively short term. I think more emphasis needs to be placed on this point in the final section of the introduction as a mosaic landscape containing kelp forest and urchin barrens is not in itself a unique situation.

The field and laboratory investigations have been well designed and executed and the methods section contains sufficient detail to replicate.

Validity of the findings

Line 83 states that 89 sites were surveyed however the associated csv file only contains data for 83 sites (assuming each site corresponds to a single row). Is the csv file missing the other six sites and were these included in the analysis or excluded for some reason?

Line 165: The mean lantern index value for both kelp forests and urchin barrens is the same (0.32) thought the statement above says they are significantly different. Is one of these values a typo?

Figure 2: The statistical values in this figure are somewhat confusing and are not elaborated on in the figure legend. The P values in the figure do not correspond to the P values that are reported between Line 168 and 171. The in figure values suggest a significant effect (P<0.001) on lantern index for both the density of Brown Algae and Folised Red Algae whereas both P values in the manuscript for these algae types show non-significance. Could you please explain what this difference is and why two seemingly different statistical outcomes are being reported?

Line 173: I don’t think a post-hoc Tukey Test is necessary here seeming that there are only two levels of the one factor (Zone = Barren or Kelp Forest). It would suffice to say that the gonad index is significantly higher in kelp forests (index value) than in urchin barrens (index value, F/P values).

Additional comments

An interesting read and tidy little experiment which shows how sea urchins can adapt to changing food availability in the relatively short term after kelp forests have changed to urchin barrens.

---

## Round 0.2 · Minor Revisions

Thank you for revising your MS.

Could you please address the following before the MS is accepted:

Line 54 (in Word doc): This is vague: "widespread kelp deforestation associated with an outbreak of purple sea urchins". Do you mean "widespread kelp deforestation followed an outbreak of purple sea urchins"?
67: Bennett misspelt
97: change "conducted" to "surveyed"
131: There's no mention in the Methods of the ANCOVA on gonad data, and some of the details on the ANCOVAs in the response to reviewer one could be included in the Methods as they relate to lanterns and/or gonads.
171: Change to "Similarly, the relationships between lantern length and lantern width, and between test diameter and test height, did not vary with habitat type (Figure 2)."
174: Change to "An analysis of covariance (ANCOVA) revealed that relative lantern length
176: Change to "Conversely, gonad condition was significantly higher in forests than in barrens"
Para starting line 179: None of the R2 values are >0.4, so are not "strong" (two instances). Use "correlated" rather than "associated"
183: Change to "Cover of encrusting algae was negatively correlated with"
217: rephrase "widely found"
313: use lower case for species & subspecies, other words in title
334: Diadema misspelt
342: Rodgers misspelt
Fig 2 legend: "test heighted"?
Fig 3: The comparisons in 3B & 3D are potentially misleading due to differences in body size between habitats, and could be deleted. Figs 3A & 3C show that lantern length and GI are both strongly correlated with body size so those plots are the appropriate way to display the data.
Fig 3C would be better as raw gonad weight vs test diameter (you might have to log both axes to get a linear relationship). Gonad index already incorporates body size (as weight) so it doesn't make sense to plot it against another measure of size (diameter).
Fig 4 legend: explain what the letters A-F on the fig panels correspond to, or preferably add the information to the title of each graph so the reader doesn't have to keep looking between the figure and the legend
Fig 4: why use different colours for the 95% CIs? If you do want to use different colours then colour-code them as per Fig 5.

·

Basic reporting

No comment

Experimental design

No comment

Validity of the findings

No comment

Additional comments

Thank you for sending through a revised manuscript and taking the time to go through each of the comments made in the initial review. The revisions made have rounded out the paper well.

---

## Round 0.3 · Minor Revisions

Thanks for making the changes requested. They all look fine except there's still two references to "strong" correlations where r2=0.4 or less (lines 211 & 212 in the pdf version). Please tone down "strong", which usually implies r2 values of at least 7 or 8.

---

## Round 0.4 · accepted · Accept

Thanks for making the requested changes to your manuscript, which I'm pleased to accept for publication in PeerJ.